# A novel deep learning-based framework with particle swarm optimisation for intrusion detection in computer networks

**Abdullah Asım Yılmaz** *

Computer Engineering Department, Atılım University, Ankara, Turkey

* abdullah.yilmaz@atilim.edu.tr

## Abstract

Intrusion detection plays a significant role in the provision of information security. The most critical element is the ability to precisely identify different types of intrusions into the network. However, the detection of intrusions poses a important challenge, as many new types of intrusion are now generated by cyber-attackers every day. A robust system is still elusive, despite the various strategies that have been proposed in recent years. Hence, a novel deep-learning-based architecture for detecting intrusions into a computer network is proposed in this paper. The aim is to construct a hybrid system that enhances the efficiency and accuracy of intrusion detection. The main contribution of our work is a novel deep learning-based hybrid architecture in which PSO is used for hyperparameter optimisation and three well-known pre-trained network models are combined in an optimised way. The suggested method involves six key stages: data gathering, pre-processing, deep neural network (DNN) architecture design, optimisation of hyperparameters, training, and evaluation of the trained DNN. To verify the superiority of the suggested method over alternative state-of-the-art schemes, it was evaluated on the KDDCUP'99, NSL-KDD and UNSW-NB15 datasets. Our empirical findings show that the proposed model successfully and correctly classifies different types of attacks with 82.44%, 90.42% and 93.55% accuracy values obtained on UNSW-B15, NSL-KDD and KDDCUP'99 datasets, respectively, and outperforms alternative schemes in the literature.

## Introduction

In the realm of network security research, network intrusion detection systems (IDSs) have great importance. Through the use of active protection technology, these systems can identify signs of intrusion and swiftly respond by taking the necessary measures to halt such intrusions. These measures may include issuing warnings to users or implementing other relevant safeguards [1]. The various intrusion detection methods in the literature primarily fall into two types: anomaly-based IDS (AIDS) and signature-based IDS (SIDS) [2]. Anomaly-based detection entails formulating a model that describes the customary behaviour of network traffic; any observed activity that diverges from this model is then deemed to be an intrusion. This type of

**Data Availability Statement:** The raw code used in this study is publicly available on GitHub at https://github.com/abasimyilmaz/DL-Based_IDS_Framework_with_PSO and Zenodo at

https://doi.org/10.5281/zenodo.11212077. The study utilized three publicly accessible datasets: the KDDCUP'99 dataset, available at https:// kdd.ics.uci.edu/databases/kddcup99/kddcup99. html; the NSL-KDD dataset, available at https:// www.kaggle.com/datasets/hassan06/nslkdd; and the UNSW-NB15 dataset, available at https://research.unsw.edu.au/projects/unsw-nb15-dataset.

**Funding:** The author(s) received no specific funding for this work.

**Competing interests:** The author have declared that no competing interests exist.

method is effective at identifying previously unknown attacks [3]. Signature-based systems operate primarily on the basis of identified intrusion attacks; in scenarios where attack signatures are unknown, the identification of anomalous network activity is predominantly accomplished through the use of anomaly-based systems [4]. The development of a strong and efficient network IDS remains a key challenge for researchers in network security. Despite considerable advancements in IDS technology, a considerable number of solutions still rely on less effective signature-based methods, rather than employing the more capable anomaly detection techniques [5]. To overcome the challenges of creating a versatile and effective network IDS to handle unforeseen and unpredictable attacks, one possible solution is to implement a deep-learning-based approach [6]. When applying machine learning (ML) in practical applications, experts in the field have typically designed the features used to describe samples, and specialised knowledge is then necessary to process the data effectively. The quality of these features plays a critical role in the model's generalisation performance, and the design of good features is a challenging and error-prone task. In contrast, deep learning techniques are specifically designed to learn superior feature representations through the analysis of vast quantities of unlabelled data, which can then be applied to the classification process [7]. Deep learning has been used for concerted efforts in this direction, and represents a distinct subfield of ML [8].

Over the past few years, significant numbers of researchers have adopted deep learning techniques across diverse fields, such as graph-based applications [9], hand gesture recognition [10], enhancement of driving safety [11, 12], identification of human actions [13], recognition of facial emotions [14], speech recognition [15], natural language processing [16], malware classification [17–19], medical applications [20], and object detection [21]. The achievements of deep learning techniques have attracted the interest of scholars in the field of intrusion detection, and research in this field has been undertaken to tackle the shortcomings of existing intrusion detection techniques and to elevate the overall effectiveness of models [22]. However, the development of a resilient system remains challenging. Robust systems are of the utmost importance for reducing feature spaces and computational complexity, surmounting the constraints imposed by time, resources, and hardware, and augmenting the accuracy rates of IDSs. Hence, this study presents a new framework for detecting intrusions that exploits the capabilities of deep learning methods. The objective is to build a hybrid system that improves intrusion detection's effectiveness and precision.

In this study, intrusion data were first collected from the KDDCUP'99 [23], NSL-KDD [24] and UNSW-NB15 [25] datasets, and were pre-processed, which involved data coding, standardisation and conversion. When the pre-processing stage was complete, the convolution layers of the proposed hybrid architecture were applied to extract high-level features from the intrusion data, and high-level features were obtained. At the parameter optimisation stage, the parameters were fine-tuned for optimal performance. Finally, classification was conducted by applying the optimised parameters to the features that were extracted. In essence, the proposed approach combines multiple extensive deep learning models, based on the transfer learning technique, to create a hybrid model. In the stages described above, the rectified linear unit (ReLU) function and several hidden layers were applied. Moreover, parameter optimisation for these deep learning algorithms was carried out with particle swarm optimisation (PSO). The outcomes of the test phase demonstrated that the proposed approach efficiently extracted distinctive features for each category of attack, thus enabling effective classification. The empirical results also indicated that the proposed deep learning method achieved superior accuracy in terms of classifying distinct attack categories compared to existing methods in the literature.

This study makes five major contributions to the literature. Firstly, it introduces a novel hybrid deep-learning-based intrusion detection method. Secondly, the PSO method is used to

optimise the hyperparameters, thus enhancing the performance of the model. Thirdly, the method is evaluated on three extensive intrusion detection datasets to ensure its effectiveness and applicability. Fourthly, the proposed approach incorporates a new hash layer that combines three pre-trained models, which distinguishes it from conventional deep learning approaches that use only one model. Lastly, the method can significantly reduce the feature space and achieves superior accuracy compared to other known methods, representing a valuable advancement in intrusion detection research.

The remaining sections of this work are structured as follows. Section 2 presents background information about intrusion detection and a concise overview of pertinent studies in this field. Section 3 provides a detailed explanation of the proposed framework. A discussion and some experimental results are given in Section 4, with an explanation of the intrusion detection datasets used. The paper is concluded with a concise summary and a discussion of potential future work.

## Related work

In this section, we present some necessary background information and review the studies of IDSs in the literature that are relevant to our study to enable the reader to gain an understanding of the concepts that underpin the model proposed in this paper.

IDSs can be classified as detection or deployment techniques, and a framework depicting this taxonomy is provided in Fig 1. An IDS can be subclassified as a SIDS or an AIDS from a detection-based perspective, whereas from a deployment- based perspective, it can be classified as a network-based IDS (NIDS), hybrid IDS, or host-based IDS (HIDS) [26–28]. Among these classifications, Machine learning-based, Deep learning-based and Hybrid IDSs, which are relevant to our study, will be explained in detail with examples in the following subsections. Table 1 summarizes various deep leraning (DL), ML, and Hybrid approaches for intrusion detection systems with various datasets. Among these classifications, Machine learning-based, Deep learning-based and Hybrid IDSs, which are relevant to our study, will be explained in detail with examples in the following subsections. In addition to the examples in the subsections, Table 1 summarizes various DL, ML, and Hybrid technique for intrusion detection systems with various datasets. The dataset utilized to test the system, the author and year of publication, and the DL, ML, and hybrid IDS techniques are listed in Table 1. When reviewing

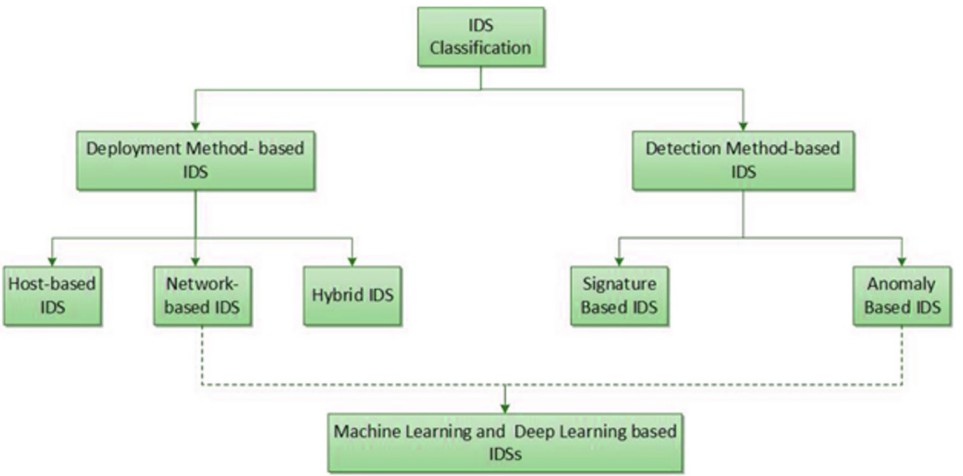

**Fig 1. Classification taxonomy for intrusion detection systems.**

**Table 1. The IDS works in the literature.**

| Paper | Year | Dataset | Technique |
|---|---|---|---|
| Ravale et al. [29] | 2015 | KDDCUP'99 | K-means, RBF |
| Al-Yaseen et al. [30] | 2017 | KDDCUP'99 | K-means, Support Vector Machine |
| Yang et al. [31] | 2018 | KDDCUP'99, NSL-KDD | Support Vector Machine, PSO |
| Farahnakian et al. [32] | 2018 | KDDCUP'99 | Deep autoencoder (DAE) |
| Gao et al. [33] | 2019 | NSL-KDD | Ensemble learning model with various ML algorithms |
| Krishna et al. [34] | 2020 | KDDCUP'99 | Multi-layer perceptron |
| Nayyar et al. [35] | 2020 | CICID2017 | Long short-term memory |
| Chen et al. [36] | 2020 | CICID2017 | Convolutional Neural Network |
| Issa et al. [37] | 2023 | NSL-KDD | Hybrid method based on LSTM and CNN |
| Qazi et al. [38] | 2023 | CICID2018 | Hybrid deep learning techniques based on Recurrent Neural Networks (RNN) and CNN |

the studies in the literature presented in Table 1, it is evident that systems integrating various DL and ML techniques have been developed. The KDDCUP'99, NSL-KDD, and CICID2017 datasets—which are often used in the literature—are used in the tests of the created systems.

## Machine learning-based IDS

ML models can be divided into two primary types, unsupervised and supervised [39], as depicted in Fig 2. In unsupervised learning, labels are not attached to the training data passed to the system. Examples of supervised learning include K-means clustering, reinforcement learning, and principal component analysis. In supervised learning, the training data passed to the algorithm contain the desired solutions, which are known as labels. Examples of supervised learning methods include decision trees (DTs), linear regression, SVM, and k-nearest neighbours (KNN). Bayesian classification, SVM and similar ML algorithms are commonly employed in the existing literature for IDSs. ML intrusion detection systems are based on five

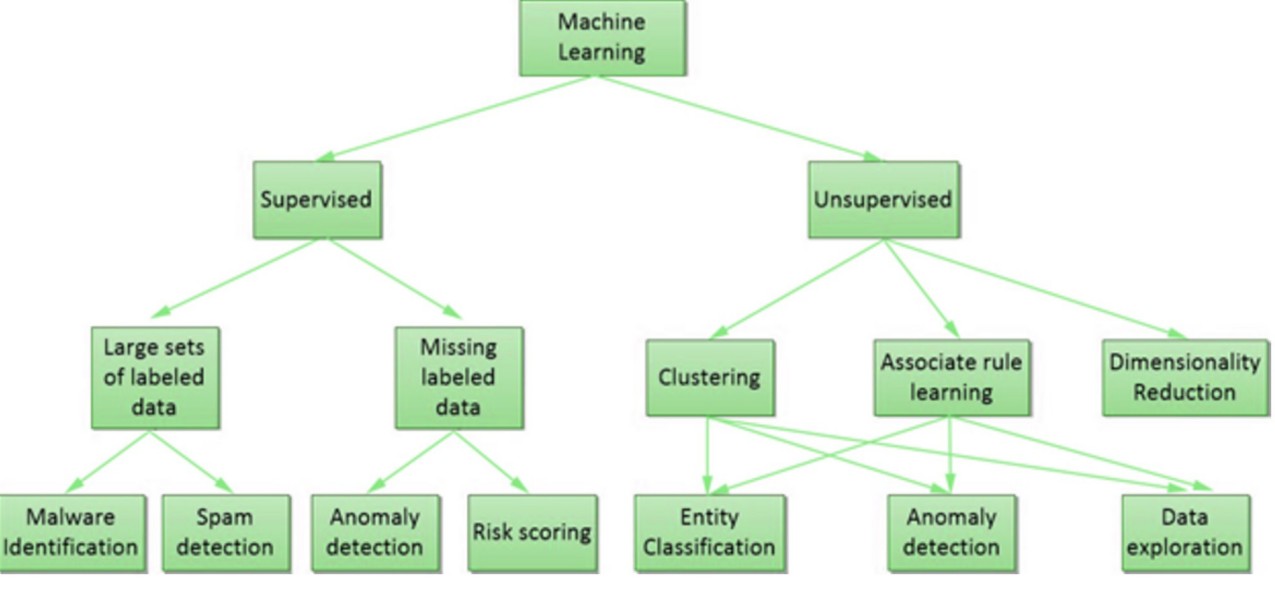

**Fig 2. ML algorithms in cyber security.**

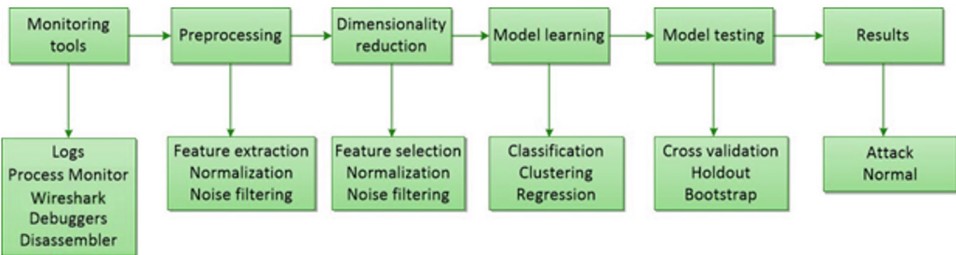

**Fig 3. Basic methodology of an ML-based intrusion detection system.**

stages, as shown in Fig 3. The initial step is to collect data using monitoring tools, and the second involves pre-processing the gathered data. In the third stage, dimensionality reduction is carried out to obtain more accurate results. The model is then trained, and finally, results are obtained by testing the trained model.

Singh et al. [40] developed an intrusion detection technique based upon the online sequential extreme learning machine. Their goal was to address the challenges faced by IDSs, such as low detection rates, high false alarm rates, and the need to handle large amounts of data. The suggested approach employed an ensemble of feature selection techniques including filtered, correlation, and consistency-based methods, in order to eliminate irrelevant features. Furthermore, this method leveraged alpha profiling to alleviate the problem of time complexity, while beta profiling was employed to reduce the size of the training dataset. The empirical results showed a detection time of 2.43 s and a false positive rate of 1.74%, with high accuracy, on the binary class NSL-KDD dataset.

Costa et al. [41] conducted research on the application of ML techniques in the context of IoT security, with a specific focus on intrusion detection. The aim of their study was to provide an overview of recent works in the field and their contributions. Various papers in the literature were reviewed, and the challenges faced in IoT environments were highlighted. The use of ML and deep learning algorithms for intrusion detection was emphasised, and their promising results were discussed. The article also mentioned the importance of addressing false positive rates in intrusion detection, while considering the trade-off between improved recognition rates and increased computational burden. Overall, these authors concluded that intrusion detection in the context of IoT remains a challenge, but highlighted the ongoing efforts in terms of developing optimised security protocols to protect data while minimising energy consumption.

A comprehensive article by Buczak and Guven [42] in 2016 surveyed various ML algorithms, such as SVM, intended for the purpose of anomaly detection. The article delved into the intricacies of ML and DL algorithms, examined the difficulties associated with employing ML and DL in the realm of cybersecurity, and put forth a number of suggestions and recommendations. In another study in the same year, Aburomman et al. [43] introduced a unique approach for constructing ensembles of classifiers (SVM and KNN) using weights generated by PSO. The authors emphasised that the use of weights generated by metaheuristic techniques could lead to enhanced accuracy in IDSs. Although SVM is considered a cutting-edge ML algorithm, its performance relies heavily on the careful selection of appropriate parameters. Enache et al. [44] introduced an IDS in which the information gain was used to select relevant features, and which included an SVM classifier. The parameters for the SVM were determined using the artificial bee colony (ABC) and PSO swarm intelligence algorithms. The experimental results showed that the optimised IDS model achieved a high accuracy rate when fine-tuned using PSO or ABC on the NSL-KDD dataset.

## Deep learning-based IDS

Deep learning models are typically created using deep neural network (DNN) structures. A DNN is an artificial neural network with many layers, and learning is performed via the connections between these layers. Deep learning models learn using multiple layers of these DNN structures, and attempts to learn the complex relationships in the dataset [45].

Recurrent neural networks (RNNs) and CNNs represent two of the most frequently used types of DNN models. CNNs are specifically designed to accept raw data as input, thereby eliminating the need for feature extraction or image reconstruction. In addition, they have a relatively low number of parameters and require only a small amount of data. Due to these advantages, CNNs have proven to be exceptionally successful in performing tasks related to image recognition [46], natural language processing and so on. In the case of certain network traffic protocols, CNNs have shown a capacity for performing well due to their ability to be trained quickly [47].

Fan and Ling-Zhi [48] achieved a high level of accuracy in feature extraction by employing a multilayer CNN architecture in which a convolution layer was linked to the sampling layer. When applied to the KDDCUP'99 dataset, this model demonstrated superior performance compared to conventional detection algorithms such as SVM. However, CNNs are unable to consider timing knowledge in a specific traffic scenario, and are limited to analysing a single input package. In a real-world attack scenario, a single packet of traffic may appear as normal data, and may be identified as malicious traffic only when a large number of packets is transmitted concurrently or within a brief time frame. CNNs are inadequate in this case, as they could lead to a large number of missed alarms. In contrast, RNNs are frequently used to analyse sequential data, and long short-term memory (LSTM), a type of RNN, has performed well in applications that require the analysis of sequential data, such as natural language processing [48] Kim et al. [49] evaluated the performance of various algorithms, including a Bayesian approach, SVM, KNN, a product-based neural network, a generalised regression neural network, and the LSTM-RNN network, on the KDDCUP'99 dataset. The LSTM-RNN network performed better than the other methods. Hassan et al. [50] introduced a combined CNN and weight-dropped LSTM (WDLSTM) deep learning approach for detecting network intrusions, and conducted experiments on the UNSW-NB15 dataset. The authors of the work employed WDLSTM to maintain long-term dependencies between the features in order to prevent overfitting in recurrent connections and CNN to derive meaningful features from the massive IDS data. On the UNSW-NB15 dataset, the performance of the suggested hybrid method was compared with that of conventional methods and it is obtained satisfactory performance.

## Hybrid IDS

Hybrid intelligent systems have been created to overcome the limitations of traditional IDSs, which often struggle with low detection rates for new attacks and a high incidence of false positive alerts. A hybrid approach involves merging both misuse-based and anomaly-based techniques to achieve a more comprehensive and effective detection capability [51]. An HIDS combines both anomaly and signature-based IDSs to strike a balance between storage and computing costs while minimising false positive alarms. This approach has become increasingly popular due to its efficient detection capabilities and ease of use [52]. An HIDS can enhance the accuracy of intrusion detection by integrating multiple detection models, and typically consists of two main components: the first component receives and processes unclassified data, while the second component analyses the processed data and identifies potential intrusion activities. The ultimate goal is to achieve superior detection performance through the combined efforts of these two components [53]. There are three main categories of HIDSs:

clustered models with a single hybrid, integrated hybrids and cascaded hybrids. Each of these categories represents a distinct approach to combining multiple detection models for improved intrusion detection capabilities [54].

Issa et al. [37] proposes a novel deep learning-based Intrusion Detection System (IDS) for detecting Distributed Denial- of-Service (DDoS) attacks by hybridizing Convolutional Neural Networks (CNN) and Long Short-Term Memory (LSTM) networks. The model leverages CNN for automatic feature extraction and LSTM for sequence prediction, creating a seven-layer architecture to enhance detection performance. The model is tested using the NSL-KDD dataset, and it achieves superior performance metrics, a high accuracy rate, which surpasses traditional CNN and LSTM models as well as other state-of-the-art approaches. The results demonstrate the model's effectiveness in improving detection accuracy and reliability for DDoS attack scenarios. Alghayadh et al. [55] developed a hybrid intrusion detection model specifically for enhancing smart home security. The model consisted of two different components, each serving a different purpose: in the first module, ML algorithms such as KNN, DT, XGBoost and random forest were used to enable real-time intrusion detection, whereas in the second module, the misuse intrusion detection technique was applied to identify known attack patterns. To evaluate the performance of this model, the researchers conducted tests using both the CSE-CIC-IDS2018 and NSL-KDD datasets. The results demonstrated the model's exceptional ability to detect network intrusion and user-based anomalies in the context of smart homes.

Saheed et al. [56] introduces a hybrid feature selection approach combining the Bat meta-heuristic algorithm with the Residue Number System (RNS) to enhance intrusion detection systems (IDS). The Bat algorithm is used to identify significant features by optimizing search spaces, while RNS improves computational speed and reduces complexity through its modular arithmetic properties. Principal Component Analysis (PCA) is applied for feature extraction, and classification is performed using Naïve Bayes (NB) and K-Nearest Neighbors (KNN) algorithms. Experimental evaluations on the NSL-KDD dataset demonstrated the superior performance of the Bat-RNS+PCA+KNN model, achieving a detection high accuracy, precision and F-Score values.

Abdulganiyu et al. [57] developed XIDINTFL-VAE model to detect minority class intrusions to detect minority class intrusions in highly imbalanced network traffic. This framework combines Class-Wise Focal Loss (CWFL) with a Variational AutoEncoder (VAE) and integrates XGBoost as the classifier. The CWFL-VAE generates synthetic data tailored to the minority classes, improving the detection capabilities for rare and critical attacks. XGBoost is used for robust classification, benefiting from its ability to handle structured data and imbalances. Experiments were conducted on the NSL-KDD and CSE-CIC-IDS2018 datasets, demonstrating significant improvements over traditional methods such as SMOTE and ADASYN, as well as existing classifiers like Logistic Regression and Random Forest. The XIDINTFL-VAE achieved high performance across metrics, with obtained high precision, F1 score and recall values, effectively balancing detection accuracy and minimizing false alarms.

Saheed et al. [58] proposes the IoT-Defender framework, a lightweight intrusion detection system (IDS) designed to enhance IoT network security using edge computing. IoT-Defender integrates a Modified Genetic Algorithm (MGA) for optimal feature selection and a Fine-Tuned Long Short-Term Memory (LSTM) model for anomaly detection. The framework employs MGA to identify the most relevant features from IoT-specific datasets (BoT-IoT, UNSW-NB15, and N-BaIoT) and uses a genetic algorithm to fine-tune the LSTM architecture, optimizing parameters like the number of hidden layers and learning rates. The system also incorporates a focal loss function to address class imbalance in IoT traffic data. Experimental results on edge devices such as the Raspberry Pi 4 show that IoT-Defender has

a 2.56% It shows that it achieves a low false alarm rate and achieves high accuracy and detection rate.

Even though a lot of research has been done on intrusion detection, it is still quite challenging to detect different kinds of network intrusions effectively in the field of cybersecurity. Cyberattacks are also growing and changing at exponential rates, and many of these modern cyberattacks cannot yet be detected by any reliable system or technique. Additionally, traditional and contemporary IDSs are no longer able to differentiate sophisticated intrusions from typical network traffic. Therefore, a novel hybrid deep learning architecture for the effective detection of many intrusion types has been developed in this study. The proposed architecture is a hybrid of three pre-trained networks based on transfer learning. Our work's primary contribution is a novel deep learning-based hybrid architecture that combines three well-known pre-trained network models in an optimized manner while using PSO for hyperparameter optimization.

## Proposed method

A novel framework for intrusion detection is introduced in this section. Our framework leverages deep learning algorithms and incorporates a hybrid DNN architecture that supports hyperparameter optimisation. The most important contribution of this framework is its unique hash architecture, which effectively combines three commonly used pre-trained network models and enables hyperparameter optimisation. The methodology of our system consists of five main phases, as shown in Fig 4. First, intrusion detection data are collected from two extensive datasets. The data are then subjected to pre-processing, as described in depth in the subsection on data pre-processing. Thirdly, high- and low-level intrusion detection features are extracted using pre-trained networks in the feature extractor phase, which also involves performing hyperparameter optimisation. At this stage, the ResNet50 [59], GoogLeNet [60] and AlexNet [61] deep learning architectures are used for feature extraction/selection process, while the PSO method is used for hyperparameter optimization. Next, the proposed architecture is trained using the optimised hyperparameters obtained through PSO together with the intrusion detection datasets in the fully connected (FC) layers. Finally, the output phase involves classification using a SoftMax classifier. To assess the performance of the proposed method, we carried out experiments on three extensive intrusion datasets. The results demonstrated that our method achieved high accuracy in terms of classifying intrusion detection data and outperformed existing state-of-the-art methods. We provide a detailed explanation of the performance results in Section 4.

The remainder of this section is structured as follows. We describe the pre-processing of the data, give an overview of the suggested model, and discuss the schemes in the literature that were used in our method. In the data pre-processing subsection, we describe how the

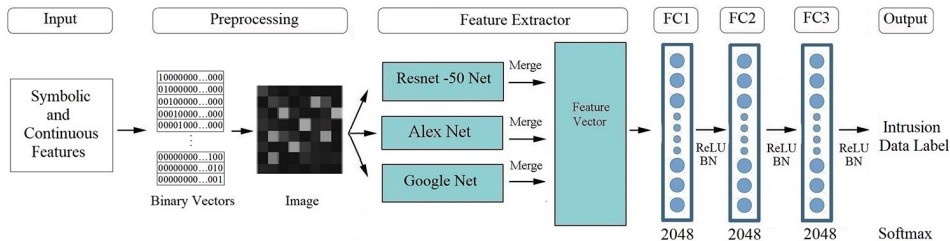

**Fig 4. Basic methodology of an ML-based intrusion detection system.**

intrusion detection data are converted into visual images for identification using a CNN. The next subsection provides a detailed description of the proposed intrusion detection framework. Lastly, we explain the deep learning architectures employed in our method and discuss the literature pertaining to PSO.

## Data pre-processing

In our scheme, a pre-processing method designed by Li et al. [62], in which a CNN is used, is applied to define an image representation of the intrusion detection data. The aim is to convert intrusion data into a visual image format. To accomplish this, different types of features are mapped into a binary vector space, which is then transformed into an image.

In the intrusion data attributes, there are three symbolic datatypes: the protocol type, flag, and service. These features are encoded using a one-hot encoder, resulting in binary vectors. For instance, the protocol type, with values icmp, udp, and tcp, is transformed into binary vectors with three dimensions, (100, 010, 100), as illustrated in Fig 5(a).

In this method, a standard scaler min-max normalisation technique is applied to manage the continuous features in intrusion data consisting of integer and floating point types. The purpose is here to transform the continuous data to the range [0, 1]. This approach can ensure that the values of the features are rescaled proportionally to fit the desired range. Min-max

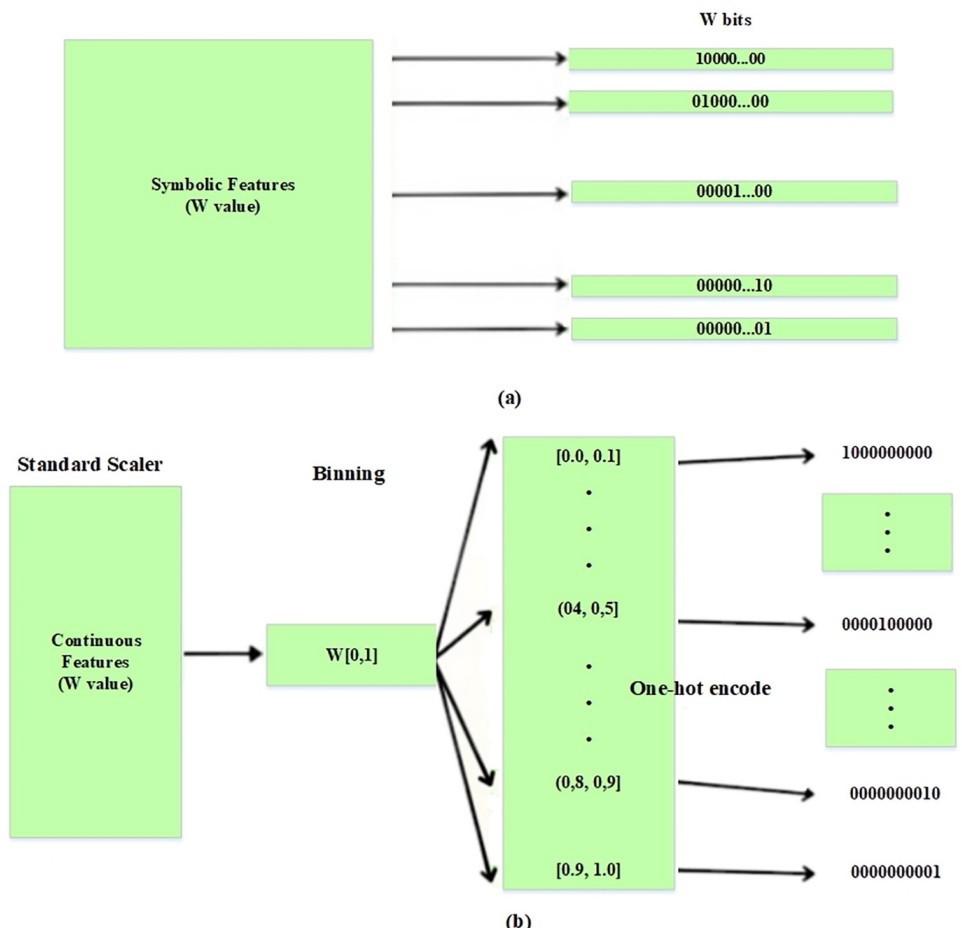

**Fig 5. (a) One-hot encoding (b) Binarisation and discretisation process for continuous features.**

normalisation was carried out using the formulation in Eq (1)):

$$w_{new} = \frac{w - w_{min}}{w_{max} - w_{min}},$$

(1)

where $w_{max}$, $w_{min}$, and $x$ represent the maximum value of the feature, the minimum value and the numeric feature value respectively, whereas x new represent value after normalisation process in equation. The scaled continuous value is divided into 0 intervals after the normalisation step. The order number of intervals are then encoded into 10 binary vectors using a one-hot encoder, as shown in Fig 5(b).

## Overview of the proposed method for intrusion detection

Our approach provides an optimised framework for the task of intrusion detection, since it is built as a hybrid DNN. The suggested structure, depicted Fig 4, comprises six phases: collection of intrusion data, pre-processing, creation of the DNN architecture, fine-tuning of the hyperparameters, training, and assessment. A flowchart of the system is shown in Fig 6 to give

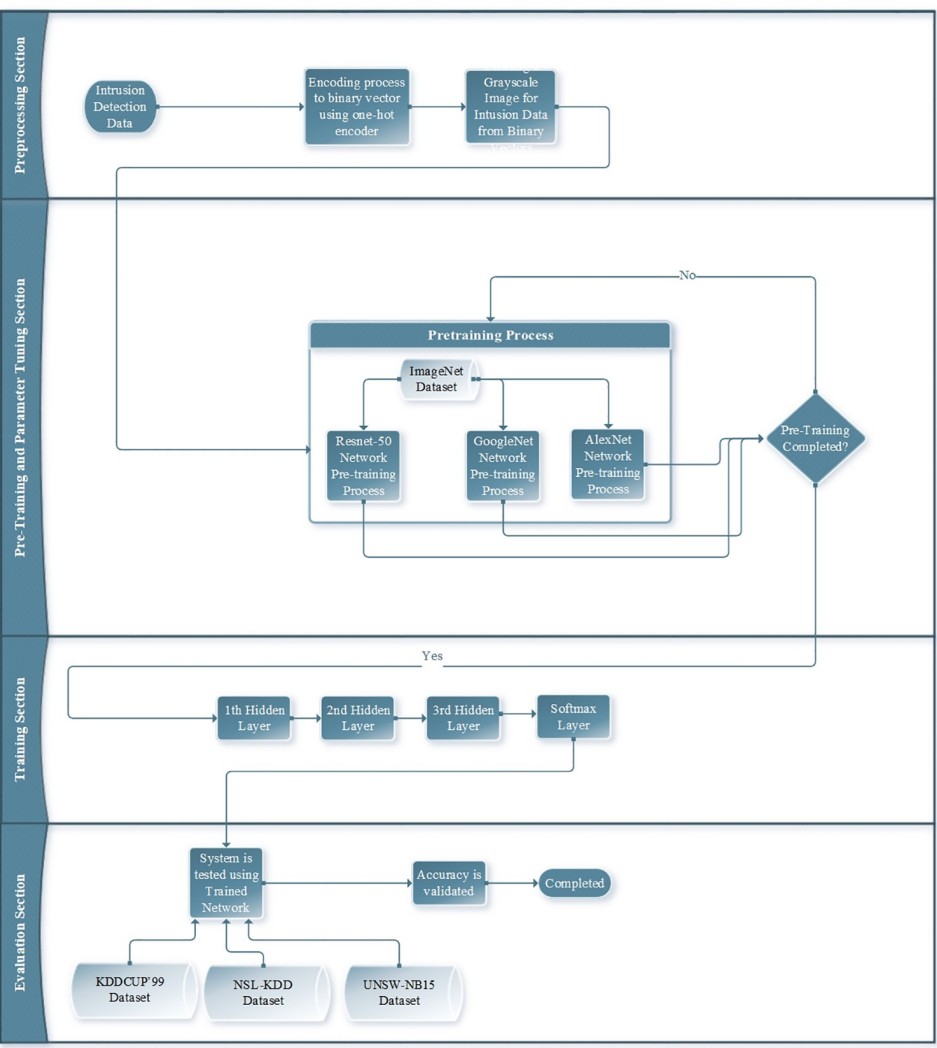

**Fig 6. Flowchart of proposed architecture for intrusion detection.**

Fig 7. Images of intrusion data samples.

a more thorough illustration of these five processes. Feature extraction is done using the pre-trained networks that were used in the parameter tuning and pre-training steps. At the training stage, the last layer is a SoftMax classifier, while the first three layers are FC layers that carry out learning operations.

Intrusion data are converted into a binary vector following this pre-processing step. The acquired binary vector is then converted into an $8 \times 8$ greyscale image with zero padding for any empty pixels. Fig 7 displays samples of visualised intrusion image frames that were produced as a result of these processes.

In the first step, samples of intrusion data are collected from the KDDCUP'99 [23], NSL-KDD [24] and UNSW-NB15 [25] datasets. A comprehensive description of these intrusion datasets is provided in Section 4. The intrusion data then undergo a pre-processing step, the specifics of which are detailed in the previous subsection.

The architecture of the suggested DNN is then created. At this point, a process is first carried out to determine an suitable deep learning architecture. A hash model is constructed by combining pre-trained architectures, since preliminary experiments indicated that a hash model produced a higher total precision [38]. This hash module includes the ResNet50, GoogLeNet and AlexNet architectures. Transfer learning is then applied. Adapting a model trained on one task to do a different but related task is known as transfer learning, and this is carried out by stripping the architecture of the pre-trained model down to a specific layer and then adding new layers that are relevant to the current problem. In this way, the medium- and low-level feature extraction layers of the pre-trained model are transferred to the new model, and high-level feature extraction is accomplished using the layers that are added to the architecture to reflect the relevant classification problem. Transfer learning methods have been extensively used to tackle challenges related to classification processes, such as large dataset sizes, model complexity, time constraints and hardware resource limits. In view of these challenges, transfer learning is adopted in the proposed architecture.

Next, our deep-learning-based architecture undergoes hyperparameter tuning in order to optimise the parameters. The automated process of adjusting deep learning or ML model hyperparameters through optimisation approaches is known as hyperparameter tuning [63]. PSO, a widely used metaheuristic optimisation method, is applied to tune the hyperparameters in our approach, as this has a low time complexity and can support a variety of hyperparameters.

After hyperparameter tuning has been carried out, training is conducted in order to achieve a high accuracy rate. In this stage, public comprehensive datasets mentioned above and the PSO method, which is a metaheuristic method and whose details are explained in the Proposed Method and Prior Schemes Used in the Suggested Method Sections, were used to minimize the problem of class imbalance in the training data and obtain high accuracy values. In addition, 20% of the intrusion data are used for testing, 10% for validation, and 70% for training.

Finally, an empirical analysis is carried out by passing comprehensive datasets as input to the trained model. A detailed explanation of the empirical analysis and the results from the model are provided in Section 4.

In summary, PSO is used for hyperparameter optimisation of the transfer learning-based model, and three pre-trained networks are combined using an equal weighting operation to produce feature vectors. The stages of the process are as follows. Firstly, intrusion data are gathered. Secondly, pre-processing is applied to the gathered data. The third step involves hyperparameter tuning, and the pre-training operation is then conducted. The ImageNet dataset [64] is used to train the Resnet50, GoogLeNet and AlexNet architectures in this step. Following this, 6144-dimensional combined feature vectors are generated by merging the features acquired from the ResNet50, AlexNet, and GoogLeNet models. These features are extracted from the final FC layers as 2048-dimensional vectors. To normalise the 6144-dimensional combined feature vectors, they are transmitted from the FC layers to the SoftMax layer. At this point, there are 18 outputs in the SoftMax layer, representing the 18 different types of intrusion, while the FC layers consist of 6144 nodes.

## Prior schemes used in the suggested method

The schemes in the literature that were used to design the proposed intrusion detection architecture are reviewed in this section. In this section, we introduce three CNN architectures and the PSO algorithms used as a hyperparameter tuning method.

**Particle swarm optimisation.** PSO is an algorithm for optimisation that draws inspiration from the collective behaviour of flocks of birds and fish schools. It aims to find the best solution to a problem based on the cooperation and communication of a group of potential solutions called particles [65] The particles in PSO represent potential solutions that move through a multidimensional search space. Each particle's velocity determines the direction and magnitude of its movement, and its position represents a solution. Particles explore the search space by adjusting their velocities and positions based upon the best experience of the entire group and their own experience [66]. The algorithm begins by initialising particles with random velocities and positions. In each iterative step, the velocities and positions of the particles are updated based upon the best position found by all particles in the group and their own best positions. This update is affected by the collective behaviour of the group and individual experiences [67]. The velocity update process in PSO consists of two main parts, known as the social and cognitive constituents. The cognitive step guides particles towards their best positions, while the social step pulls them towards the best position found by any particle in the group. These components strike a balance between exploitation and exploration, allowing for effective searches of the solution space [68]. The PSO algorithm repeats until a termination criterion is met, such as convergence to a solution, reaching a desired fitness value or reaching a maximum number of iterations. The final positions of the particles reflect optimal solutions, or the best solution discovered [69].

**CNN arhitectures.** ResNet-50 [59] is a CNN with a depth of 50 layers. The goal of the ResNet model is to resolve the problem of a decrease in performance with CNNs, which is done by adding shortcuts called residual learning blocks between the layers. These blocks, illustrated in Fig 8(a), can be considered the core structural components of ResNet. In a residual block, the path that results from adding the input x directly to the network's output is referred to as a shortcut or jump link. The ResNet50 architecture, as shown in Fig 8(a), has 25.6 million parameters, and includes a SoftMax layer, two pooling operations, an FC layer and five convolutional blocks.

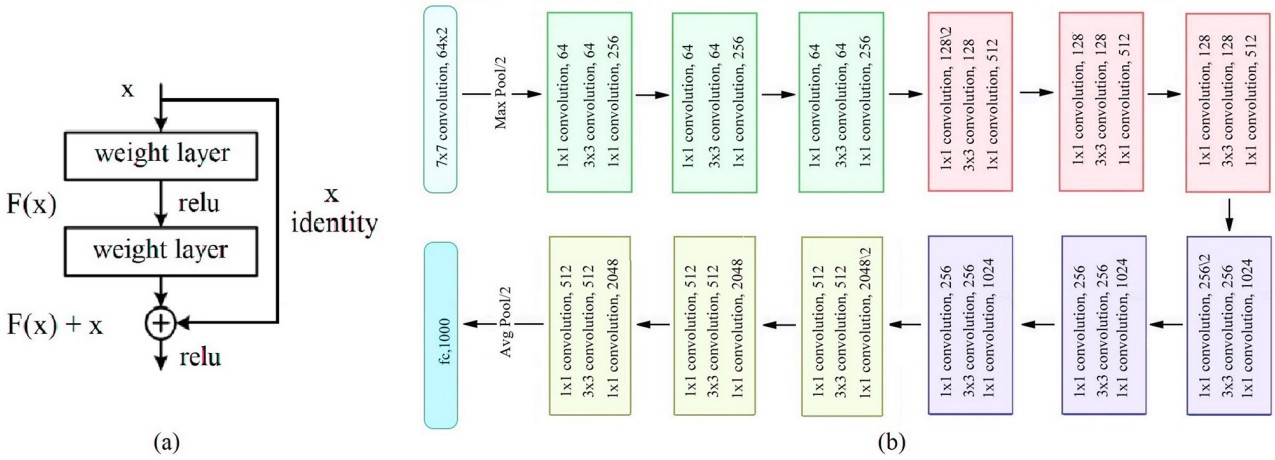

**Fig 8. (a) Residual block and (b) basic architecture of the ResNet50 model [59].**

GoogLeNet [60] is a CNN with seven million process parameters and a depth of 27 layers, including pooling layers. It is also known as Inception, and is a deep CNN architecture developed by researchers at Google. It includes Inception modules, which use multiple convolutional filter sizes in parallel to capture features at different scales. This module consists of a shortcut branch and few deeper branches, and it provides the width item in the model to be obtained. The architecture of GoogLeNet [57], illustrated in Fig 9, aims to strike a balance between depth and computational efficiency by reducing the number of parameters. The architecture consists of four main sections: the output classifier, stacked, stem, and Inception auxiliary classifiers modules. In addition, this architectur e has 144 layers, including FC, convolution, max-pooling, output, SoftMax, input layers and ReLUs, and it contains nine Inception modules.

AlexNet [61] is a groundbreaking deep CNN architecture that played a crucial role in popularising deep learning for computer vision tasks. Developed by Alex Krizhevsky et al., it achieved remarkable success by winning the ImageNet Large Scale Visual Recognition Challenge in 2012 [70] This network, as shown in Fig 10, consists of seven ReLUs, two normalisation layers and three pooling layers following the convolution layers. In addition, it includes a SoftMax layer and FC layers for classification and learning.

## Experimental results and discussion

The results of our experiments, an evaluation of the proposed model, the details of the implementation and a description of the datasets are presented in this section. The Python scripting language was used to implement our model, and experiments were performed in a Linux environment on a personal computer with 128 GB of RAM and Intel Core i9 12950HX processor with a speed of 5.2 GHz. The test, training and validation data were chosen randomly from the datasets, and evaluation operations were carried out separately. A total of 10% of the data were used for validation, 20% for testing, and 70% for training. Without GPU support, the training process was carried out for about 50 hours and stopped at 100 epochs. The hyperparameters selected for the experiments are shown in Table 2.

### Benchmark datasets

The popular datasets that were used to test the performance of the proposed method are described in detail in this subsection. A detailed summary of the datasets and the classes of

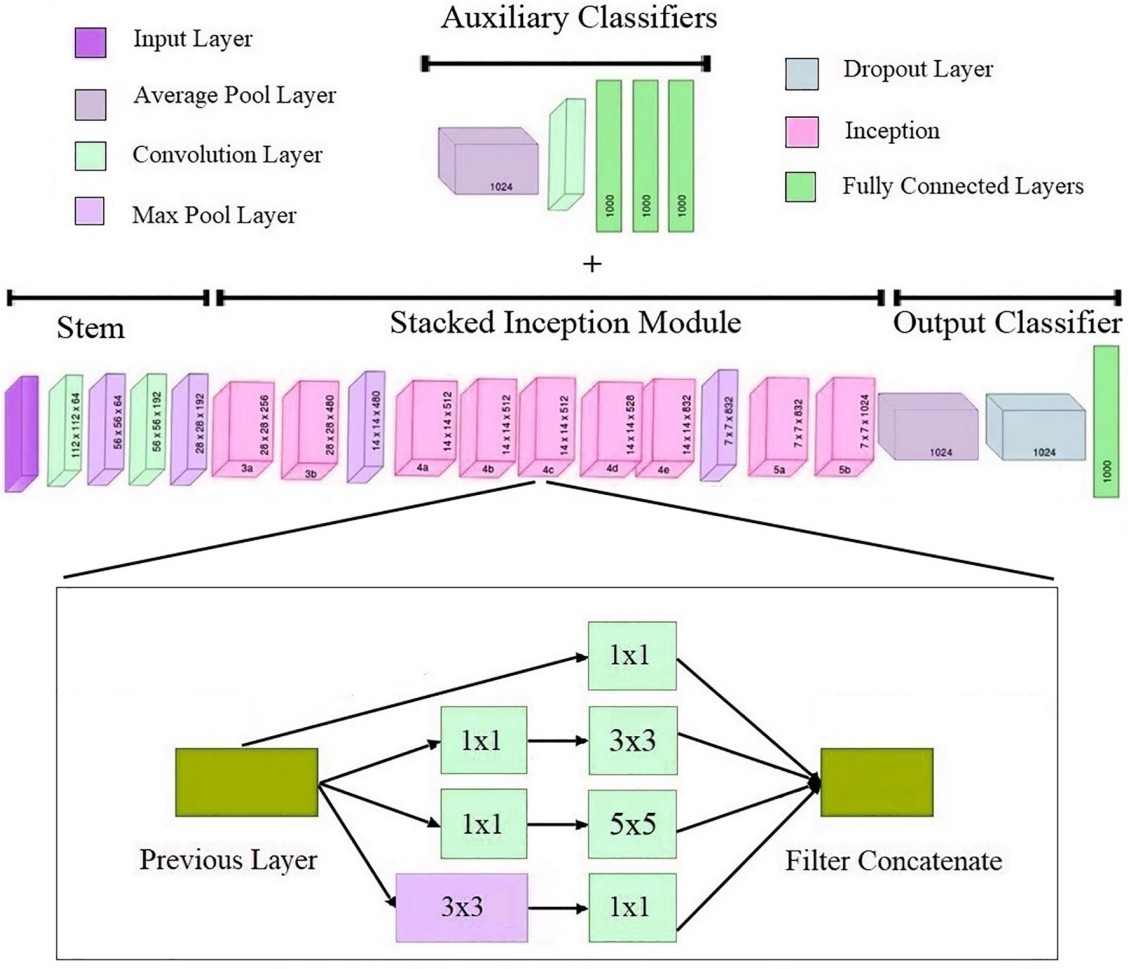

**Fig 9. Basic architecture of the GoogLeNet model [60].**

attack within each is given in Table 3. The details of the testing and training datasets used for the experiments are displayed in Tables 4 and 5.

**KDDCUP'99 dataset.** KDDCUP'99 [23] is a widely used benchmark dataset for intrusion detection research. It was created as part of the 1999 Knowledge Discovery and Data Mining

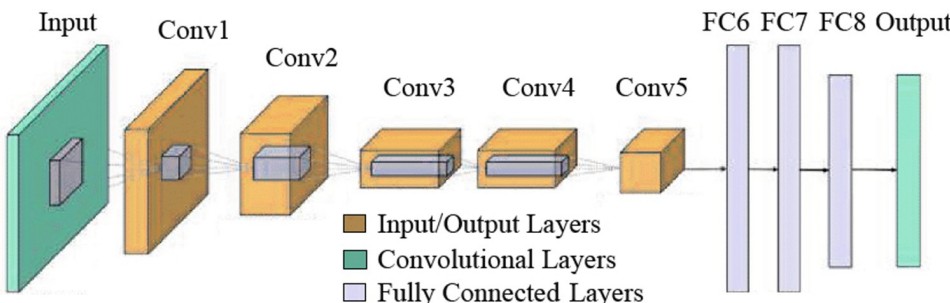

**Fig 10. Basic architecture of the AlexNet model [61].**

**Table 2. Hyperparameters of the suggested model.**

| Hyperparameter | Value(s) |
|---|---|
| Epoch Count | 100 |
| Loss Type | Categorical cross entropy |
| Optimizer | PSO |
| Learning rate | 0.0001 |
| Dropout | 0.5 |
| Batch size | 64 |
| Validation, test, train split ratios | 0.1:0.2:0.7 |

**Table 3. Summary of public benchmark datasets.**

| Name | Quantity | Attack Type Category |
|---|---|---|
| KDDCUP'99 [23] | 4 | U2R, R2L, probe, DoS |
| NSL-KDD [24] | 4 | U2R, R2L, probe, DoS |
| UNSW-NB15 [25] | 9 | Worms, shellcode, reconnaissance, fuzzers, exploits, port scans, DoS, generic, backdoors |

**Table 4. Composition of 10% KDDCUP'99 and NSL-KDD datasets.**

| Class of attack | Name of attack | Data instances—10% data | | | |
|---|---|---|---|---|---|
| | | KDDCUP'99 | | NSL-KDD | |
| | | Test | Train | Test | Train |
| DOS | 'pod' | 87 | 264 | 41 | 201 |
| | 'Land' | 9 | 21 | 7 | 18 |
| | 'teardrop' | 12 | 979 | 12 | 892 |
| | 'neptnue' | 58001 | 107201 | 4657 | 41214 |
| | 'back' | 1098 | 2203 | 359 | 956 |
| | 'smurf' | 164091 | 280790 | 665 | 2646 |
| Probe | 'portsweep' | 354 | 1040 | 157 | 2931 |
| | 'nmap' | 84 | 231 | 73 | 1493 |
| | 'satan' | 1633 | 1589 | 735 | 3633 |
| | 'ipsweep' | 306 | 1247 | 141 | 3599 |
| R2L | 'spy' | 0 | 2 | 0 | 2 |
| | 'warezmaster' | 1602 | 20 | 944 | 20 |
| | 'imap' | 1 | 12 | 1 | 11 |
| | 'warezclient' | 0 | 1020 | 0 | 890 |
| | 'phf' | 2 | 4 | 2 | 4 |
| | 'ftpwrite' | 3 | 8 | 3 | 8 |
| | 'guess_password' | 4367 | 53 | 1231 | 53 |
| | 'multihop' | 18 | 7 | 18 | 7 |
| U2R | 'multihop' | 18 | 7 | 18 | 7 |
| | 'perl' | 2 | 3 | 2 | 3 |
| | 'buffer overflow' | 22 | 30 | 20 | 30 |
| | 'rootkit' | 13 | 10 | 13 | 10 |
| | 'loadmodule' | 2 | 9 | 2 | 9 |
| Normal | | 60593 | 97278 | 9711 | 67343 |
| Total | | 311029 | 494021 | 22544 | 125973 |

**Table 5. Distributions of samples in the UNSW-NB15 dataset used for training.**

| Class of attack | Size |
|---|---|
| DoS | 12264 |
| Shellcode | 1133 |
| Generic | 40000 |
| Exploits | 33393 |
| Worms | 130 |
| Fuzzers | 18184 |
| Normal | 56000 |
| Analysis | 2000 |
| Backdoor | 1746 |
| Reconnaissance | 10491 |
| Total | 175341 |

(KDD) Cup competition organised by the National Science Foundation and DARPA. This dataset contains a large number of network traffic records, both normal and malicious, which are used to train and evaluate IDSs [70]. In dataset, each record is characterized by 41 parameters and labeled either as normal traffic or an attack of a specific type.

The attacks in this dataset are classified into four primary categories: U2R (user to root), R2L (root to local), probe (probing attacks), DoS (denial of service). The training dataset also includes 24 specific types of intrusions with an additional 14 attacks in the testing dataset, which includes the aforementioned four main types of attack.

**NSL-KDD dataset.** NSL-KDD [24] is a publicly available dataset that was created by building upon the KDDCUP'99 dataset. Important problems that can significantly impact the accuracy of intrusion detection and lead to a false assessment of AIDS were identified through a statistical analysis of the KDDCUP'99 dataset, with large numbers of duplicate packets being the primary issue. The creators of the NSL-KDD dataset addressed these problems in the KDD-CUP dataset by removing duplicate records from the training and testing sets and increasing the proportion of minority samples in the testing set. As discussed above, the KDDCUP'99 dataset has 41 features, and intrusion attacks are categorised into four types: U2R, R2L, Probe, and DoS.

**UNSW-NB15 dataset.** UNSW-NB15 [25] was produced by the Australian Centre for Cyber Security, and includes roughly two million records with 49 features, extracted using Bro-IDS, Argus tools and some newly developed algorithms. This dataset includes nine types of intrusion attack: exploits, shellcode, DoS, reconnaissance, backdoor, generic, fuzzers, worms and port scans.

## Results and discussion

The performance of DNN models is assessed based on evaluation metrics that are of crucial importance when evaluating classification processes. These metrics distinguish between model results and measure the performance of the classification model [71]. The classification performance of the proposed method was assessed using the F1-score, accuracy, specificity, sensitivity, the area under the curve of a receiver operating characteristic (ROC-AUC) metrics, and the results are presented in this subsection. The formulae in Table 6 were used to calculate these metrics, where TN and FN refer to true negative and false negative, respectively, while TP and FP refer to true positive and false positive.

**Table 6. The evaluation metrics formulas.**

| Evaluation Metric | Equation |
|---|---|
| Specificity | TN / (TN+FP) |
| Accuracy | (TP+TN) / (TP+TN+FP+FN) |
| Sensitivity | TP / (TP+FN) |
| F-score | 2*TP / (2*TP+FP+FN) |

The results for the sensitivity, specificity, F1-score, and accuracy metrics on the individual datasets for the ResNet152, GoogLeNet, and AlexNet models as well as the suggested network are shown in Fig 11. It can be seen that our model performs better than the other three DNN models. In addition, whereas the performances of the three DNNs differ significantly on the three datasets, our model produces similar performance results for all three, showing that our network outperforms the other three deep neural networks in terms of performance and robustness.

Following the models' training over 100 epochs, the ROC-AUC curves were plotted, as seen in Fig 12, respectively. Plotting the true positive rate (TPR) against the false positive rate (FPR) at various classification thresholds yields the ROC-AUC curve. Test ROC-AUC values for the three models were 96.6%, 96.9%, and 97.4%, respectively.

Confusion matrices were also used to investigate the performance for various types of intrusion attacks. The confusion matrices for five intrusion attack types (U2R, R2L, Probe, DoS and normal traffic) of the ResNet50, GoogLeNet, AlexNet and proposed deep neural network models are shown in Fig 13 for the NSL-KDD dataset. Here, accuracy rates are shown for each type of intrusion attack through the use of confusion matrices. It can be seen from Fig 13(d) that the proposed approach produces better results than the other models for all types of intrusion apart from U2R. In addition, compared to the other network models, ResNet50, shown in

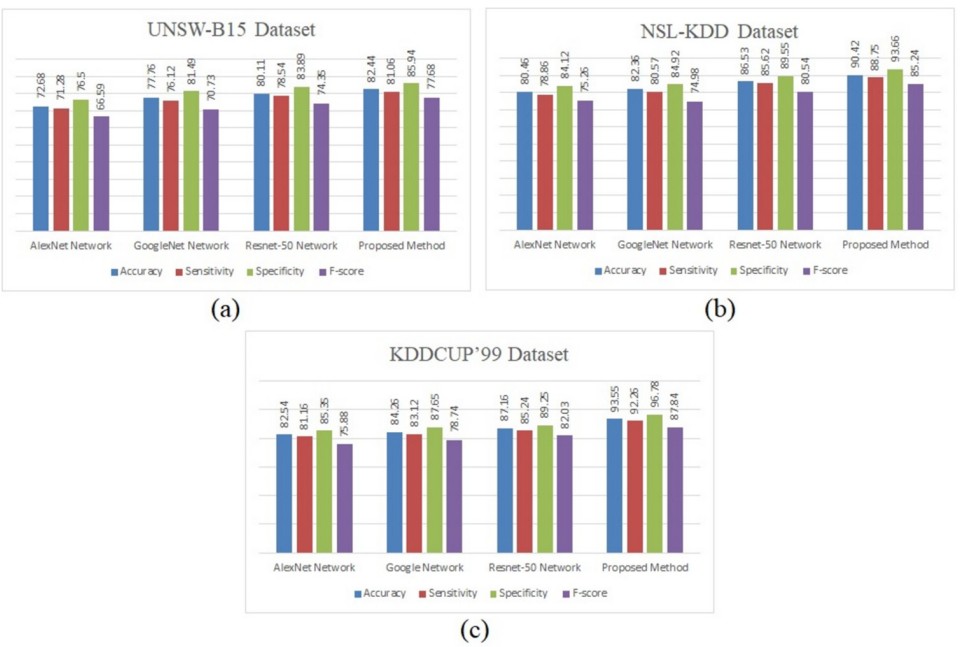

**Fig 11. Quantitative results for the (a) UNSW-B15, (b) NSL-KDD, and (c) KDDCUP'99 datasets.**

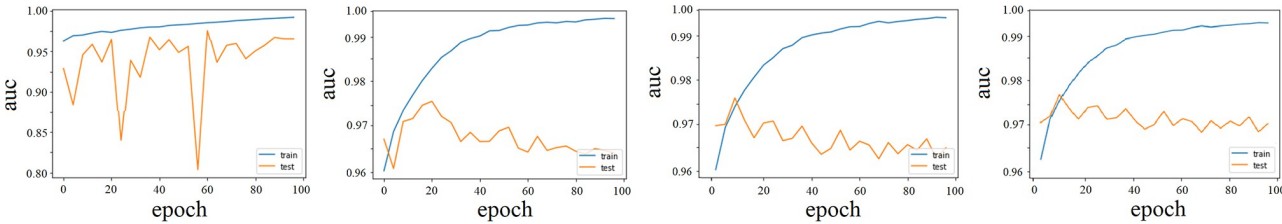

**Fig 12. ROC-AUC of the four models (a) AlexNet (b) GoogleNet (c)Resnet-50 (d) Proposed Model.**

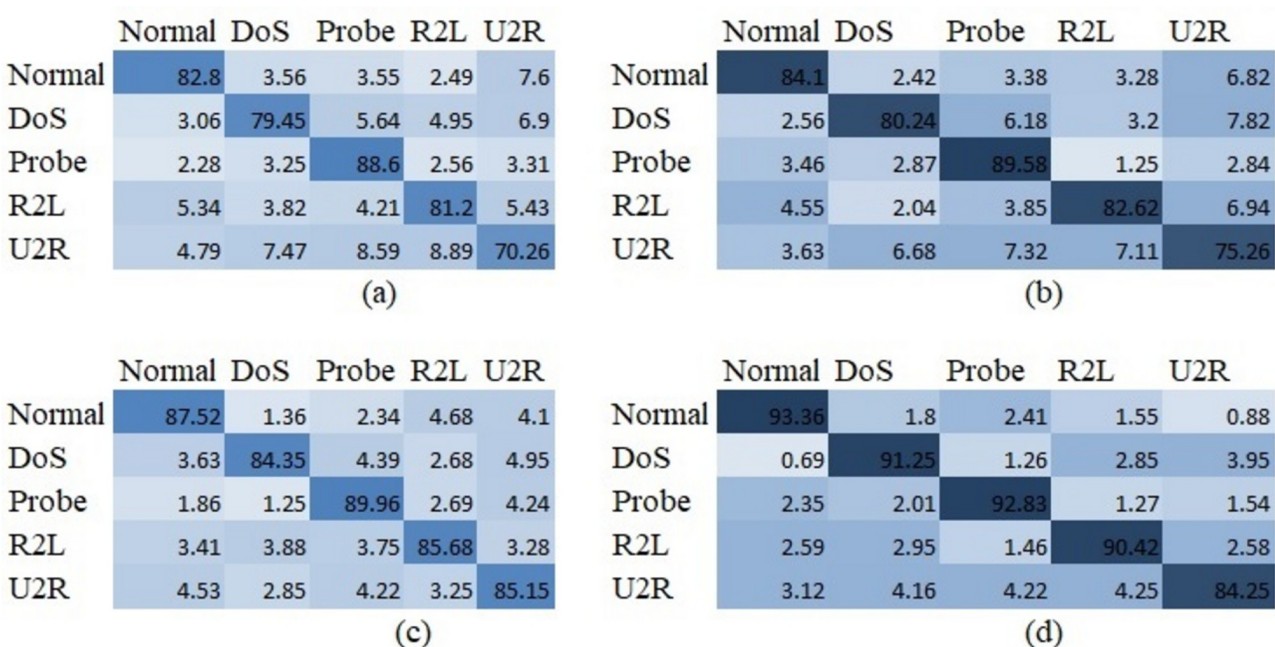

**Fig 13. Confusion matrices obtained by applying the proposed method to the NSL-KDD dataset, for five types of intrusion.**

Fig 7(c), achieves superior detection of the U2R attack. In this case, the Probe and R2L intrusion attack types were readily detected by each network.

Finally, a comparison was performed against state-of-the-art intrusion detection methods in order to assess the performance of our approach. The values for accuracy obtained on KDD CUP'99, NSL-KDD and UNSW-NB15 datasets for these state-of- the-art schemes and the proposed model are given in Table 7. We note that the suggested model is more accurate and efficient than the alternatives based on the higher value of accuracy.

## Conclusion

Effective detection of various types of network intrusion continues to be an extremely difficult task in the area of cyber-security, despite the fact that a great deal of research has been done on intrusion detection. Cyberattacks are also increasing and evolving at exponential rates, and there is currently no method that can identify many of these contemporary cyber-related attacks. Complicated cyberattacks can no longer be distinguished from normal network traffic by traditional IDSs. Hence, this paper has presented a novel deep learning architecture for the

**Table 7. The comparison of results from the proposed architecture and state-of-the-art algorithms on the UNSW-B15, NSL-KDD and KDDCUP'99 dataset.**

| Study | Model/Methodology | Accuracy |
|---|---|---|
| **Intrusion detection accuracy results for UNSW-B15** | | |
| **Study** | **Model/Methodology** | **Accuracy** |
| Vinyakumar et al. [72] | Naive Bayes | 43.7 |
| Vinyakumar et al. [72] | Logistic Regression | 53.8 |
| Vinyakumar et al. [72] | Deep Neural Network | 65.1 |
| Vinyakumar et al. [72] | Decision Tree | 73.3 |
| Vinyakumar et al. [73] | Decision Tree | 75.4 |
| **Proposed Method** | **Proposed Architecture+PSO** | **82.44** |
| **Intrusion detection accuracy results for NSL-KDD** | | |
| **Study** | **Model/Methodology** | **Accuracy** |
| Vinyakumar et al. [72] | Naive Bayes | 29.5 |
| Vinyakumar et al. [72] | Logistic Regression | 61.2 |
| Vinyakumar et al. [73] | Multi-layer Perceptron | 72.1 |
| Vinyakumar et al. [72] | Decision Tree | 76.3 |
| Li et al. [74] | GoogleNet | 77.04 |
| Vinyakumar et al. [73] | Integer-Only Deep Recurrent Neural Networks | 78.3 |
| Vinyakumar et al. [72] | Deep Neural Network | 78.5 |
| Li et al. [74] | Resnet50 | 79.14 |
| Yin et al. [75] | Recurrent Neural Networks | 81.29 |
| Vinyakumar et al. [73] | Gated Recurrent Unit | 82.8 |
| Vinyakumar et al. [73] | Recurrent Neural Network | 83.0 |
| Vinyakumar et al. [73] | Long Short Term Memory | 89.6 |
| **Proposed Method** | **Proposed Architecture+PSO** | **90.42** |
| **Intrusion detection accuracy results for KDDCUP'99** | | |
| **Study** | **Model/Methodology** | **Accuracy** |
| Vinyakumar et al. [72] | Logistic Regression | 80.1 |
| Vinyakumar et al. [72] | Naive Bayes | 85.7 |
| Vinyakumar et al. [72] | SVM | 89.5 |
| Ogundokun et al. [76] | Decision Tree+PSO | 89.6 |
| Vinyakumar et al. [72] | Decision Tree | 92.4 |
| Vinyakumar et al. [72] | Deep Neural Network | 92.5 |
| **Proposed Method** | **Proposed Architecture+PSO** | **93.55** |

efficient detection of several types of intrusion. The proposed architecture is a hybrid of three pre-trained networks based on transfer learning. Initially, intrusion data were gathered from three exhaustive datasets, and were pre-processed and subjected to data coding, standardisation and conversion processes. Next, high-level features were obtained using pre-trained networks. Hyperparameter adjustment was then carried out using PSO. Finally, training of the model was carried out using a supervised learning method.

Our work's primary contribution is a novel deep learning-based hybrid architecture that combines three well-known pre-trained network models in an optimized manner while using PSO for hyperparameter optimization. The proposed deep learning technique was assessed on UNSW-B15, NSL-KDD and KDDCUP'99 datasets, and the performance of the proposed hybrid model was initially compared with each model in isolation. The findings demonstrated that our approach could successfully classify intrusion with high values for the F1-score, accuracy, recall and precision metrics. Next, our model was evaluated on the NSL-KDD dataset using confusion matrices, and it was shown to generate better results than the alternative

models for all types of intrusion apart from U2R. Finally, a comparison between our architecture and state-of-the-art alternatives was performed, and results revealed that the proposed method was superior to the other methods. In future work, we plan to compare the performances of more models on different databases and employ other metaheuristic techniques for hyperparameter optimisation.

## Acknowledgments

I would like to express our deepest gratitude to my mother Serap YILMAZ and my wife Süheyla YILMAZ for their unwavering support and help.

## Author Contributions

**Conceptualization:** Abdullah Asım Yılmaz.

**Data curation:** Abdullah Asım Yılmaz.

**Formal analysis:** Abdullah Asım Yılmaz.

**Funding acquisition:** Abdullah Asım Yılmaz.

**Investigation:** Abdullah Asım Yılmaz.

**Methodology:** Abdullah Asım Yılmaz.

**Project administration:** Abdullah Asım Yılmaz.

**Resources:** Abdullah Asım Yılmaz.

**Software:** Abdullah Asım Yılmaz.

**Supervision:** Abdullah Asım Yılmaz.

**Validation:** Abdullah Asım Yılmaz.

**Visualization:** Abdullah Asım Yılmaz.

**Writing – original draft:** Abdullah Asım Yılmaz.

**Writing – review & editing:** Abdullah Asım Yılmaz.

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
