## [Decision Letter · Decision Letter 0]

23 Nov 2024

PONE-D-24-38218A novel deep learning-based framework with particle swarm optimisation for intrusion detection in computer networksPLOS ONE

Dear Dr. YILMAZ,

Thank you for submitting your manuscript to PLOS ONE. After careful consideration, we feel that it has merit but does not fully meet PLOS ONE’s publication criteria as it currently stands. Therefore, we invite you to submit a revised version of the manuscript that addresses the points raised during the review process.

We look forward to receiving your revised manuscript.

Kind regards,

Raman Singh

Academic Editor

PLOS ONE

Journal Requirements:

2. We note you have included a table to which you do not refer in the text of your manuscript. Please ensure that you refer to Table 2 in your text; if accepted, production will need this reference to link the reader to the Table.

3. Please remove your figures from within your manuscript file, leaving only the individual TIFF/EPS image files, uploaded separately. These will be automatically included in the reviewers’ PDF.

Reviewers' comments:

Reviewer's Responses to Questions

**Comments to the Author**

1. Is the manuscript technically sound, and do the data support the conclusions?

Reviewer #1: Yes

2. Has the statistical analysis been performed appropriately and rigorously? 

Reviewer #1: Yes

3. Have the authors made all data underlying the findings in their manuscript fully available?

Reviewer #1: Yes

4. Is the manuscript presented in an intelligible fashion and written in standard English?

Reviewer #1: Yes

5. Review Comments to the Author

Reviewer #1: Reviewer Comments on a novel deep learning-based framework with particle swarm optimisation for intrusion detection in computer networks

1. The motivation and contribution of this paper should be stated more clearly in the abstract to better understand from the beginning of the study. Authors are advised to be precise in the abstract, and structure your abstract as follows- 1) Background 2) Aim/Objective 3) Methodology 4) Results 5) Conclusion. Write 2-4 lines for each and merge everything in one paragraph (150-250 Words) without any subheading.

2. What are the main contributions of the paper in terms of novelty, technique used and observation? Since there are many Hybrid-Intrusion detection system available in literature, what are the issues this paper address is missing in the paper.

3. Authors should explain the feature selection algorithms used in the work.

4. The paper does not explicitly address the issue of class imbalance in the training data. Imbalanced datasets can lead to biased models, and it's important to understand how the proposed research handles this issue.

5. The related work section is very small, an updated and complete literature review should be conducted. Some latest papers listed below which studied similar problems can be discussed: 1). Feature selection in intrusion detection systems: a new hybrid fusion of Bat algorithm and Residue Number System; 2). XIDINTFL‑VAE: XGBoost‑based intrusion detection of imbalance network traffic via class‑wise focal loss variational autoencoder.

3). Modified genetic algorithm and fine-tuned long short-term memory network for intrusion detection in the internet of things networks with edge capabilities.

6. In the end of related work section, highlight in 10-15 lines what overall technical gaps are observed in existing techniques that led to the design of the proposed methodology.

7. Although the English is generally quite good, there are quite a few minor grammatical errors, and a careful read-through is needed to eliminate these errors. The spelling mistake should be corrected by reading through the manuscript.

6. PLOS authors have the option to publish the peer review history of their article (what does this mean?). If published, this will include your full peer review and any attached files.

Reviewer #1: **Yes: **Yakub Kayode Saheed

---

## [Author Response · Author response to Decision Letter 0]

5 Dec 2024

Dear Editor and Reviewers,

We would like to thank you for allowing a resubmission of our manuscript, with an opportunity to address the reviewers’ comments. We also would like to thank the reviewers for their constructive comments to improve the current presentation of the paper.

We are uploading (a) our point-by-point response to the comments (response to reviewers), (b) A marked-up copy of your manuscript that highlights changes made to the original version (revised manuscript with track changes), and (c) a clean updated/revised manuscript without tracked changes (manuscript).

Best regards,

Abdullah Asım YILMAZ

---

## [Decision Letter · Decision Letter 1]

10 Dec 2024

A novel deep learning-based framework with particle swarm optimisation for intrusion detection in computer networks

PONE-D-24-38218R1

Dear Dr. YILMAZ,

We’re pleased to inform you that your manuscript has been judged scientifically suitable for publication and will be formally accepted for publication once it meets all outstanding technical requirements.

Kind regards,

Raman Singh

Academic Editor

PLOS ONE

Additional Editor Comments (optional):

Reviewers' comments:

Reviewer's Responses to Questions

**Comments to the Author**

1. If the authors have adequately addressed your comments raised in a previous round of review and you feel that this manuscript is now acceptable for publication, you may indicate that here to bypass the “Comments to the Author” section, enter your conflict of interest statement in the “Confidential to Editor” section, and submit your "Accept" recommendation.

Reviewer #1: All comments have been addressed

2. Is the manuscript technically sound, and do the data support the conclusions?

Reviewer #1: Yes

3. Has the statistical analysis been performed appropriately and rigorously? 

Reviewer #1: Yes

4. Have the authors made all data underlying the findings in their manuscript fully available?

Reviewer #1: Yes

5. Is the manuscript presented in an intelligible fashion and written in standard English?

Reviewer #1: Yes

6. Review Comments to the Author

Reviewer #1: The authors have addressed all the comments raised.

The authors have addressed all the comments raised.

7. PLOS authors have the option to publish the peer review history of their article (what does this mean?). If published, this will include your full peer review and any attached files.

Reviewer #1: No

---

## [Editor Report · Acceptance letter]

13 Dec 2024

PONE-D-24-38218R1 

PLOS ONE

Dear Dr. Yilmaz, 

I'm pleased to inform you that your manuscript has been deemed suitable for publication in PLOS ONE. Congratulations! Your manuscript is now being handed over to our production team.

Kind regards, 

on behalf of

Dr. Raman Singh 

Academic Editor

PLOS ONE